# COMPARING HUMAN AND MACHINE BIAS IN FACE RECOGNITION

## ABSTRACT

Much recent research has uncovered and discussed serious concerns of bias in facial analysis technologies, finding performance disparities between groups of people based on perceived gender, skin type, lighting condition, etc. These audits are immensely important and successful at measuring algorithmic bias but have two major challenges: the audits (1) use facial recognition datasets which lack quality metadata, like LFW and CelebA, and (2) do not compare their observed algorithmic bias to the biases of their human alternatives. In this paper, we release improvements to the LFW and CelebA datasets which will enable future researchers to obtain measurements of algorithmic bias that are not tainted by major flaws in the dataset (e.g. identical images appearing in both the gallery and test set). We also use these new data to develop a series of challenging facial identification and verification questions that we administered to various algorithms and a large, balanced sample of human reviewers. We find that both computer models and human survey participants perform significantly better at the verification task, generally obtain lower accuracy rates on dark-skinned or female subjects for both tasks, and obtain higher accuracy rates when their demographics match that of the question. Academic models exhibit comparable levels of gender bias to humans, but are significantly more biased against darker skin types than humans.

## 1 INTRODUCTION

Facial analysis systems have been the topic of intense research for decades, and instantiations of their deployment have been criticized in recent years for their intrusive privacy concerns and differential treatment of various demographic groups. Companies and governments have deployed facial recognition systems (Derringer, 2019; Hartzog, 2020; Weise & Singer, 2020) which have a wide variety of applications from relatively mundane, e.g., improved search through personal photos (Google, 2021), to rather controversial, e.g., target identification in warzones (Marson & Forrest, 2021). A flashpoint issue for facial analysis systems is their potential for biased results by demographics (Garvie, 2016; Lohr, 2018; Buolamwini & Gebru, 2018; Grother et al., 2019; Dooley et al., 2021), which make facial recognition systems controversial for socially important applications, such as use in law enforcement or the criminal justice system. To make things worse, many studies of machine bias in face recognition use datasets which themselves are imbalanced or riddled with errors, resulting in inaccurate measurements of machine bias.

It is now widely accepted that computers perform as well as or better than humans on a variety of facial recognition tasks (Lu & Tang, 2015; Grother et al., 2019) in terms of *accuracy*, but what about *bias*? The algorithm's superior overall performance, as well as speed to inference, makes the use of facial recognition technologies widely appealing in many domain areas and comes at enhanced costs to those surveilled, monitored, or targeted by their use (Lewis, 2019; Kostka et al., 2021). Many previous studies which examine and critique these technologies through algorithmic audits do so only up to the point of the algorithm's biases. They stop short of comparing these biases to that of their human alternatives. In this study, we question how the bias of the algorithm compares to human bias in order to fill in one of the largest omissions in the facial recognition bias literature.

We investigate these questions by creating a dataset through extensive hand curation which improves upon previous facial recognition bias auditing datasets, using images from two common facial recognition datasets (Huang et al., 2008; Liu et al., 2015) and fixing many of the imbalances and

erroneous labels. Common academic datasets contain many flaws that make them unacceptable for this purpose. For example, they contain many duplicate image pairs that differ only in their compression scheme or cropping. As a result, it is quite common for an image to appear in both the gallery and test set when evaluating image models, which distorts accuracy statistics when evaluating on either humans or machines. Standard datasets also contain many incorrect labels and low quality images, the prevalence of which may be unequal across different demographic groups.

We also create a survey instrument that we administer to a sample of non-expert human participants ($n = 545$) and ask machine models (both through academically trained models and commercial APIs) the same survey questions. In comparing the results of these two modalities, we conclude that, first, humans and academic models both perform better on questions with male subjects. Second, humans and academic models both perform better on questions with light-skinned subjects. Third, humans perform better on questions where the subject looks like they do. Fourth, commercial APIs are phenomenally accurate at facial recognition and we could not evaluate any major disparities in their performance across racial or gender lines. Finally, overall we found that academic models exhibit comparable levels of gender bias to humans, but are significantly more biased against darker skin types than humans.

## 2    BACKGROUND AND PRIOR WORK

We provide a brief overview of facial recognition and additional related work. We further detail similar comparative studies which contrast the performance of humans and machines. Much of the discussion of bias overlaps with the sub-field of machine learning that focuses on social and societal harms. We refer the reader to Chouldechova & Roth (2018) and Barocas et al. (2019) for additional background of that broader ecosystem and discussion around bias in machine learning.

**Facial Recognition**    In this overview, we focus on a review of the types of facial recognition technology rather than contrasting different implementations thereof. Within facial recognition, there are two large categories of tasks: verification and identification. Verification asks a 1-to-1 question: is the person in the source image the same person as in the target image? Identification asks a 1-to-many question: given the person in the source image, where does the person appear within a gallery composed of many target identities and their associated images, if at all? Modern facial recognition algorithms, such as He et al. (2016); Chen et al. (2018); Wang et al. (2018) and Deng et al. (2019), use deep neural networks to extract feature representations of faces and then compare those to match individuals. An overview of recent research on these topics can be found in Wang & Deng (2018). Other types of facial analysis technology include face detection, gender or age estimation, and facial expression recognition.

**Bias in Facial Recognition**    Bias has been studied in facial recognition for the past decade. Early work, like that of Klare et al. (2012) and O'Toole et al. (2012), focused on single-demographic effects (specifically, race and gender), whereas the more recent work of Buolamwini & Gebru (2018) uncovers unequal performance from an intersectional perspective, specifically between gender and skin tone. The latter work has been and continues to be hugely impactful both within academia and at the industry level. For example, the 2019 update to NIST FRVT specifically focused on demographic mistreatment from commercial platforms (Grother et al., 2019).

While our work focuses on the identification and comparison of bias, existing work on remedying the ills of socially impactful technology and unfair systems can be split into three (or, arguably, four (Savani et al., 2020)) focus areas: pre-, in-, and post-processing. Pre-processing work largely focuses on dataset curation and preprocessing (e.g., Feldman et al., 2015; Ryu et al., 2018; Quadrianto et al., 2019; Wang & Deng, 2020). In-processing often constrains the ML training method or optimization algorithm itself (e.g., Zafar et al., 2017a;b; Agarwal et al., 2018; Donini et al., 2018; Goel et al., 2018; Zafar et al., 2019; Diana et al., 2020; Lahoti et al., 2020; Martinez et al., 2020; Padala & Gujar, 2020; Wang & Deng, 2020), or focuses explicitly on so-called fair representation learning (e.g., Dwork et al., 2012; Zemel et al., 2013; Edwards & Storkey, 2016; Beutel et al., 2017; Madras et al., 2018; Wang et al., 2019; Adeli et al., 2021). Post-processing techniques adjust decisioning at inference time to align with fairness definitions (e.g., Hardt et al., 2016; Wang et al., 2020).

**Human Performance Comparisons**    No work in the past to our knowledge has specifically focused on the question of comparing bias or disparity between humans and machines. Some prior work

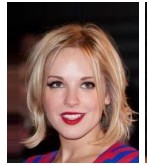 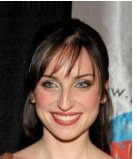 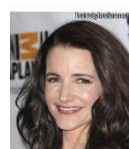 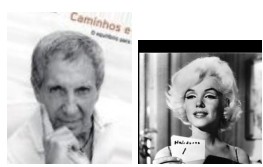

(a) Incorrect Identities: these are labeled as the same; but the left is Zoë Lister and the right is Zoe Lister-Jones.

(b) Incorrect Labelling: this individual was labeled as not being pale skinned.

(c) Black and White Images: some identities only have black and white photos.

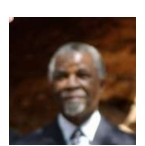 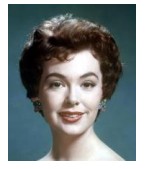 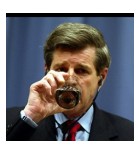 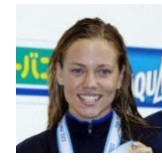 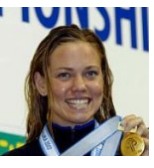 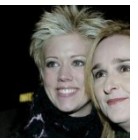

(d) Low-Quality Images

(e) Identical Attire/Background

(f) Multiple Distinct Faces

Figure 1: Shortcomings present in existing facial identification datasets

has looked at comparing overall performance or accuracy between the two groups. Tang & Wang (2004); O'Toole et al. (2007); Phillips & O'toole (2014) compare human and computer-based face verification performance. Lu & Tang (2015) was the first paper to show machine accuracy outpacing human accuracy. Hu et al. (2017); Phillips et al. (2018); Robertson et al. (2016) compared face recognition performance of human specific sub-populations whereas White et al. (2015) looked at comparing overall performance of humans who use the *outputs* of face recognition systems.

## 3 INTERRACE DATASET CURATION

We endeavor to answer two research questions: **(RQ1)** How and to what extent do humans exhibit bias in their accuracy in facial recognition tasks? **(RQ2)** How does this compare to machine learning-based models? In order to answer these questions, we created a set of challenging identification and verification questions which we posed to humans and machines from a novel dataset called InterRace for its application in intersectional facial recognition. The protocol around those experiments are described in Section 4.

To create our dataset, we first ensured that we had accurately labeled and balanced metadata. This required us to hand-check all the labels in the dataset. After removing poor quality and redundant images, we found that LFW lacked identities with dark skin tones, which is why further identities were drawn from CelebA. Though LFW does have an errata page, CelebA and other facial recognition datasets are known to have many missing or incomplete metadata, and so all CelebA images were examined by an author of this paper before adding them to the dataset. Finally, after randomly generating survey questions, we hand checked that there were no questions for which the answer is apparent or unclear for reasons other than properties of the faces (see Figure 1). In this section we detail our findings about the shortcomings in the metadata labels from LFW and CelebA and outline the steps we took to rectify and supplement these in the creation of the InterRace identities.

### 3.1 THE SHORTCOMINGS OF PREVIOUS DATASETS

In the process of trying to create a reasonable set of identification and verification questions, we identified that the LFW and CelebA datasets generally suffer from a range of problems that distort accuracy and bias metrics. We summarized these problems in Figure 1.

The first challenge we had to overcome is **incorrect identities**; this includes incorrect names, duplicated identities, as well as clearly incorrect matching between image and name. This problem is particularly harmful for facial recognition models which would be provided with galleries containing incorrect information about identities. In some cases, identities were split across multiple labels due to spellings. We found that this happened almost exclusively with non-canonically western names. E.g., Mesut Ozil (labelled as "Mesut Zil"), Jithan Ramesh (labelled as "Githan Ramesh"), Isha Koppikhar (labelled as "Eesha Koppikhar"), etc. Examples of incorrect identity labels include

Neela Rasgotra, a fictional character played by Parminder Singh and "All That Remains," a band name with the pictured individual being Philip Labonte. In other cases, multiple distinct identities were merged into the same label. In CelebA, Jennifer Lopez was grouped with Jennifer Driver, and Zoë Lister and Zoe Lister-Jones were both listed under "Zoe Lister" (pictured in Figure 1a).

Additionally, these datasets exhibit **metadata labelling problems** that manifest in two ways: (1) clearly defined labels being incorrectly or non-uniformly applied, and (2) vague and sometimes harmful metadata. In the first category, CelebA has features such as gender and age which often are incorrect or mislabeled (i.e. a pale-skinned person being labelled as not having pale skin, Figure 1b). Further, many categories in CelebA are subjective and/or harmful. For example, there is a label for "Attractive," "Big Nose/Lips," or "Chubby."

We found that some identities have **exclusively black and white images** (Figure 1c), making it trivial to identity two photos as being of the same label.

We filtered out **low-quality images** that could not be easily identified for reasons beyond properties of the face, such as poor light exposure, blurriness, facial obstruction, etc. We also removed "old-timey" photos that were easily associated with a specific time period, as this makes it easy to match them with other similar photos.

We found that many questions could be answered without considering face features at all, and these were removed. For example if the subject is **wearing identical attire and/or standing in front of an identical background in two images**. Many identities contained multiple images from the same red carpet event or award reception (Figure 1e). It *very* often happens that the same image appears multiple times in the dataset, but with slightly different crops, compression, or contrast adjustments.

Finally, some images **contained multiple faces**. Some of these pictures clearly have one person in the foreground and are therefore not problematic, but in others this is not the case, creating ambiguity as to which person is the target individual. See Figure 1f.

The image types above create inaccuracies when evaluating face recognition systems and distort measurements of bias when these problems occur at rates that differ across groups. For this reason, many datasets designed for training face analysis systems are not appropriate for evaluating bias.

## 3.2    THE INTERRACE IDENTITIES

After a thorough review of the LFW and CelebA datasets, random generation of survey questions, and rigorous hand-checking of questions to remove irregularities, we obtained a battery of survey questions for evaluating both humans and machines. We also selected survey questions that were balanced across gender, age, and skin type. Since LFW is highly skewed towards lighter identities, we included CelebA images and identities as well. We selected identities from LFW with at least two images of an individual, and then we hand labeled each identity for the following: their (1) birth date, (2) country of origin, (3) gender presentation, and (4) Fitzpatrick skin type. Labels 1-3 were assigned by an author of this paper, then that label was checked by at least two others, and modifications were made to achieve agreement among the labelers. Skin type labels (4) were assigned by 8 raters, and the mode was used as the final label.

We note that part of this work does reify categories of gender and skin type that have broader social and political implications. Further, we undertook a task of labeling and categorizing individuals who we do not know and have not received consent from for this task. Every identity for which we created these labels is indeed a celebrity in the public space with Wikipedia entries. Gender labels were rendered from the celebrity's public comments on their own gender identity.

The **Fitzpatrick scale** (Fitzpatrick, 1988) was used to help balance the survey to include subjects with diverse skin types. This scale is widely used to classify skin complexions into 6 categories. While the Fitzpatrick scale is not perfect, it is the best systematic option currently for ensuring a broad representation.

We looked up each celebrity's **birth date** online, mostly citing Wikipedia, and if we could not find it there, we continued to search on other websites. However, if we could still not find an individual's date of birth, we did not list it. To find an individual's **country of origin**, we again cited Wikipedia. If the individual came from a country that no longer existed (i.e. East and West Germany), we listed the current country. To label a person's **gender presentation**, we took note of the person's preferred

pronouns online and in interviews. In the event that their pronouns were not available online, we labeled their gender presentation. A major limitation of the CelebA and LFW datasets is that there were no individuals in our process who identified outside the gender binary or as gender queer.

At the end of our data collection, we collected metadata on 2545 identities which comprised a total of 7447 images. The identities themselves are rather imbalanced, though we selected a subgroup from these identities to create a balanced survey, discussed in Section 4. There are 1744 lighter-skinned individuals (as defined by Fitzpatrick skin types I-III) and 801 darker-skinned individuals (skin types IV-VI). There are 1660 males and 885 females. This sample is an improvement over previous datasets as it has been extensively evaluated to remove any errors in labeling and has a robust labeling for a wider array of skin types, unlike previous datasets which chose to label individuals as "pale." These data have a range of potential future use cases, such as being used for more evaluative facial recognition studies and commercial system audits.

# 4 EXPERIMENTS

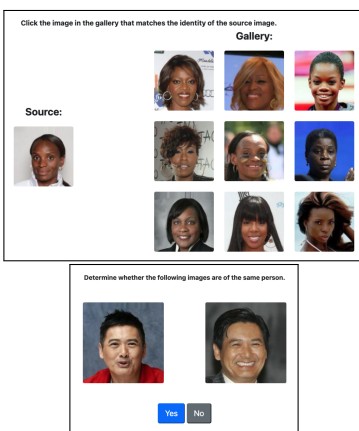

Figure 2: Example questions from the InterRace question bank. (Top) An example of an identification question. (Bottom) An example of a verification question. Notice that the demographics of all identities appearing in a question are matched to ensure the questions are not trivial.

With the high-quality metadata provided in the InterRace identities, we conduct two experiments that aim to answer our main research questions regarding the performance disparities of humans and machines. In this section, we outline how we selected the survey questions, administered the survey to human participants, and evaluated machine models. We describe the results in Section 5.

For both experiments, we create two types of questions: **identification** and **verification**. Both tasks contain a "source" image. In the identification task, 9 other images are presented in a grid, with one being of the same identity as the source and the others being of the same gender and skin type. For the verification task, a second image is selected with equal probability of being the same identity as the source image, or some other of the same gender and skin type as the source. Examples of these two types of questions can be seen in Figure 2.

We generated a static question bank with 78 identification questions and 78 verification questions for each of the 12 combinations of gender of skin type. Of those demographics with more than 78 identities, the source identity for the 78 questions were randomly chosen without replacement. This provided a total of 936 questions for each task. Finally, a pass was done over all questions to remove any for which context around the face (e.g., background or clothes) could be used to identify a person (e.g., a verification question where both images feature the same sports jersey). This final set has 901 identification and 905 verification questions.

## 4.1 HUMAN EXPERIMENT

We conducted an institutional review board-approved survey through the crowdsource platform Cint. The survey had two parts, one for each type of question: identification and verification.

Each respondent was asked 36 identification questions and 72 verification questions, for a target survey length of around 10 minutes. The questions for each user were randomly sampled from the total question bank such that an even distribution of questions were asked for each demographic group. As such, each respondent was asked 3 identification questions and 6 verification questions for each intersectional demographic identity. When the user first entered the survey they were prompted with a consent form. After completing both tasks, respondents filled out a demographic self-identification form which asked the participants their age range, gender, and skin type. When asking respondents

to evaluate their own Fitzpatrick skin type scale, we provided a brief description of the scale and respondents were also shown three examples of each skin type from our dataset. The entire text of the survey, including the demographic questions, can be seen in Appendix D.

Within each task, an attention check question was presented after the first five questions and before the last five. For the identification task, the attention check questions used an identical image for the target and in the gallery. For verification, one question consisted of pairing a light skinned female with a dark skin male (obvious negative example), and the other contained two identical images (obvious positive). The images used in these questions do not appear elsewhere in the survey. If a user failed to answer an attention check question correctly, they were screened out and any of their responses were ignored in our analysis. Additionally, any user who passed the attention checks but took fewer than 4 minutes to complete the survey was dropped from the final analysis. The first 3 verification and identification questions seen by each user were removed, to account for the possibility that the user may have taken some time to adjust to the format of the questions.

Our survey sampled English-speaking participants who were 18 years or older and were US residents. Our final sample includes 545 participants. There are 146 self-identified as dark-skinned (Fitzpatrick IV-VI) females, 128 light-skinned (Fitzpatrick I-III) females, 140 dark-skinned males, and 131 light-skinned males. Most respondents (375) came from the 20-39 and 40-59 age demographics.

Table 1: Demographic breakdown of human survey respondents used in final analysis.

|  | Fitzpatrick | Age 0-19 | Age 20-39 | Age 40-59 | Age 60-79 | Age 80+ | Total |
|---|---|---|---|---|---|---|---|
| Male | I-II | 0 | 23 | 37 | 33 | 2 | 95 |
|  | III-IV | 1 | 35 | 18 | 24 | 1 | 79 |
|  | V-VI | 4 | 43 | 33 | 17 | 0 | 97 |
| Female | I-II | 0 | 31 | 26 | 36 | 0 | 93 |
|  | III-IV | 4 | 33 | 26 | 27 | 0 | 90 |
|  | V-VI | 1 | 43 | 27 | 20 | 0 | 91 |

## 4.2 MACHINE EXPERIMENTS

**Academic Models**    To measure disparities, we trained 6 face recognition models and evaluated them on InterRace questions. We trained ResNet-18, ResNet-50 (He et al., 2016) and MobileFaceNet (Chen et al., 2018) neural networks with CosFace (Wang et al., 2018) and ArcFace (Deng et al., 2019) heads, which are designed to improve angular separation of the learned features. The models are trained using the focal loss Lin et al. (2017). For the training data, we used $140\,000$ images of 7866 CelebA identities disjoint from identities selected for the InterRace dataset. The training data has equal number of female and male identities and images. At inference time, the models solve identification questions by finding the closest gallery image in the angular feature space. For verification questions, we threshold the cosine similarity between features extracted from images in the pair.

We trained neural networks for 100 epochs with a batch size of 512 using SGD optimizer with the initial learning rate of 0.1, momentum of 0.9 and weight decay of $5e-4$. The learning rate was decreased by a factor of 10 at epochs 35, 65 and 95. Prior to training, we aligned and re-scaled training images to $112 \times 112$, during the training we randomly apply horizontal flip to images. For data pre-processing and training routines we adapt the code from the publicly available GitHub repository `face.evoLVe.PyTorch`.

**Commercial Models**    We evaluated three commercial APIs: AWS Rekognition, Microsoft Azure, and Megvii Face++. We were able to evaluate face verification and identification on AWS and Azure, and only face verification on Face++. The AWS CompareFace function, which compares a source and target image, was used for both identification and verification; the target image for identification was one image comprised of the nine gallery images stitched together. Azure has native identification and verification built into their Cognitive Services Face API. Face++ has a similar set up to AWS, however they only compare the largest detected faces in the source and target images; thus we were only able to perform face verification.

## 4.3 ANALYSIS STRATEGY

We use a two-tailed $t$-test with matched pairs (with a given pair corresponding to a single respondent's or computer model's scores on the two sections) to compare the accuracy rates between tasks. We also use two-tailed, unpaired $t$-tests to compare the overall accuracy of humans on verification questions

Table 2: Overall gender and skin type disparities exhibited by the human survey respondents, academic models, and commercial APIs.

| | Identification | | | | Verification | | | |
| | Lighter | | Darker | | Lighter | | Darker | |
| | Female | Male | Female | Male | Female | Male | Female | Male |
|---|---|---|---|---|---|---|---|---|
| Human | 67.2% | 78.3% | 55.5% | 73.1% | 78.7% | 83.1% | 73.4% | 80.1% |
| Academic Models | 95.0% | 97.5% | 89.0% | 93.6% | 96.2% | 96.7% | 92.0% | 94.2% |
| Commercial Models | 96.7% | 98.7% | 96.7% | 97.6% | 97.6% | 98.9% | 97.8% | 99.9% |

with the overall accuracy of computer models on verification questions, and the overall accuracy of humans on identification questions with the overall accuracy of computer models on identification questions. The latter $t$-tests and all $t$-tests referred to in the rest of this section are conducted on the question-level: for instance, when comparing the verification accuracy of humans and machines, we use all verification responses from all human test-takers as one sample, and all verification responses from all machines as the other.

We then analyze the disparity along gender and skin-type categories within our computer algorithms and human survey results. Users and question subjects are binned by skin type. Since the Fitzpatrick is heavily skewed towards Western conceptions of skin tone, we use two categorizations: a binary categorization of "lighter" (I-III) and "darker" (IV-VI); and categorization by (I-II), (III-IV) and (V-VI). We use two-tailed unpaired $t$-tests to detect the presence of accuracy disparities based on the gender or Fitzpatrick type of the identities that formed the questions. We perform tests of this kind on data from the six individual computer models, and also on the aggregate data sets of all human question responses and all computer algorithm responses.

We use logistic regression in our analysis to allow us to control for confounding variables. Results are reported as odds ratios, which compare the ratio of odds for a baseline event with the odds for a different event. We consider a main model for human subjects which predicts whether an individual question taken by a respondent was answered correctly, with independent variables as the question target gender and skin-type, and test-taker age, gender, and skin-type. The logistic regressions we run on the computer model responses are similar, but do not include test-taker demographics. We do report separate results for different architectures.

## 5    RESULTS

Humans achieved higher accuracy on verification (78.9%) than identification (68.3%, significant with a two-tailed matched-pair $t$-test with $p < 0.001$). For computer models as a whole, this gap persists but is substantially narrowed – performance on verification is 94.6%, with 93.7% on identification – and is no longer statistically significant ($p = 0.129$).

The performance difference between machines and humans is highly significant ($p < 0.001$) on both tasks using unpaired $t$-tests which explore group-level changes between the two tasks. Furthermore, even when controlling for demographic effects in a logistic model, humans have a much lower odds compared to computers of getting a question right (OR $= 0.14$ for verification, $p < 0.001$, OR $= 0.21$ for identification, $p < 0.001$).

**Humans and Computers Perform Better on Male Subjects**    For identification questions, we do not observe statistically significant performance for MobileFaceNetArcFace ($p = 0.09714$ for MobileFaceNetCosFace, and $p = 0.05629$ for ResNet50CosFace), but we do observe statistically significant disparities in favor of males for each of the other three ResNet models (all $p < 0.025$). In logistic regression, we observe an odds ratio for computer models on male identification subjects of 1.89 ($p < 0.001$). Similarly, humans have significantly ($p < 0.001$) better accuracy on identification questions with male subjects: 75.7% on male subjects versus 61.4% on female subjects. The same holds true for humans on verification questions: they attain an accuracy of 81.6% on male subjects, versus 76.1% on female subjects ($p < 0.001$). Interestingly, all demographics of survey respondents (when grouped by gender and skin-type) perform substantially better on males than on females for each task. The results of the human-only logistic models confirm human biases towards male subjects in both verification (OR $= 1.39$, $p < 0.001$) and identification (OR $= 1.97$, $p < 0.001$).

Academic models are found, through logistic regression, to exhibit a statistically significant difference in performance between verification questions with male or female subjects (OR $= 1.37$, $p = 0.01$).

**Humans and Computers Perform Worse on Darker-Skinned Subjects**   Humans collectively are proportionally $5.2\%$ worse on dark-skinned subjects than light-skinned subjects for verification questions ($80.9\%$ versus $76.7\%$, $p < 0.001$) when we aggregate the Fitzpatrick scale as binary. On identification questions, this proportional difference grew to $11.7\%$ in favor of light-skinned subjects ($72.7\%$ versus $64.2\%$, $p < 0.001$). This holds even when controlling for the demographics of the respondent: the odds ratio of dark-skinned compared to light-skinned question subjects for verification is $0.78$ ($p < 0.001$) while for identification it is $0.67$ ($p < 0.001$). When we aggregate the Fitzpatrick scale as three groups, I-II, III-VI, and V-VI, verification logistic regression finds statistically significant biases in favor of Fitzpatrick types I-II, over both III-VI and V-VI questions compared (OR $= 0.93$, $p = 0.023$ for III-VI; OR $= 0.85$, $p < 0.001$ for V-VI). For the identification task, even when controlling for respondent demographic, question subjects with Fitzpatrick values I-II have higher correct responses than that of values III-VI and V-VI (OR $= 0.92$, $p = 0.04$ for III-VI; OR $= 0.70$, $p < 0.001$ for V-VI).

On machines, we observe a similar higher performance on lighter-skinned subjects. When we aggregate the Fitzpatrick scale as just "light" and "dark", we observe a statistically significant performance disparity of $3.7\%$ in favor of light-skinned question subjects on the verification task ($p < 0.001$), and for identification, we observe a $5.2\%$ disparity in favor of light-skinned question subjects ($p < 0.001$). When we aggregate the Fitzpatrick scale into three categories, I-II, III-IV, and V-VI, we see a disparity for both tasks between the lightest (I-II) and darkest groups (V-VI) ($p < 0.0041$ and $p = 0.04$ for both verification and identification). Academic model performance is revealed to be significantly different, even when controlling for gender, between the types I-II and V-VI (OR $= 0.36$, $p < 0.001$ for identification; OR $= 0.47$, $p < 0.001$ for verification). However, I-II and III-VI do not show statistically significant differences for academically-trained models (OR $= 0.94$, $p = 0.714$ for identification; OR $= 0.92$, $p = 0.642$ for verification).

**Human Test-Takers Perform Better on Subjects of Similar Demographic**   We hypothesized that humans would be more accurate on questions that contained subjects that looked like them. We find evidence to support this hypothesis in our data. On the verification task, humans perform significantly better on questions where the subjects match their gender identity ($1.1\%$, $p = 0.02$), skin type ($1.7\%$, $p = 0.002$), and gender identity and skin type ($1.3\%$, $p = 0.009$). On the identification task, humans perform significantly better on questions where subjects match their skin type ($2.3\%$, $p = 0.011$) and both their gender identity and skin type ($3.3\%$, $p < 0.001$).

**Humans and Machines Exhibit Comparable Gender Disparity, but Machines have Greater Skin Type Disparity than Humans**   To test for whether the levels of disparity described above are comparable between humans and machines, we look at the confidence intervals for the odds ratios of comparable models. For both tasks, recall that we observed a disparity on gender and skin type for humans and machines. For verification, we observe that the magnitude of the gender disparities are similar (OR $95\%$ confidence intervals for humans are $[1.33, 1.46]$ and for academic models are $[1.07, 1.73]$). For identification, we observe that the magnitude of the gender disparities are also similar (OR $95\%$ confidence intervals for humans are $[1.84, 2.10]$ and for academic models are $[1.50, 2.39]$). This allows us to conclude that when there is a gender disparity displayed by both humans and machines, the magnitudes and directions of that disparity are statistically similar.

On the other hand, the skin type disparity is more pronounced in academic models than in humans. Using the same analysis technique as above, we see that the OR $95\%$ confidence intervals do not match for the darkest skin types in all cases (identification and verification as well as 2 and 3 skin type categories). Furthermore, the academically trained machines show a larger disparity (smaller odds ratio) than humans do. In identification, we have confidence intervals for binary skin types as $[0.62, 0.71]$ for humans and $[0.32, 0.52]$ for academic models. For tertiary skin types, we have confidence intervals of $[0.64, 0.75]$ for humans and $[0.27, 0.48]$ for the darkest subjects. In verification, we have confidence intervals for binary skin types as $[0.74, 0.82]$ for humans and $[0.38, 0.63]$ for academic models. For tertiary skin types, we have confidence intervals of $[0.80, 0.90]$ for humans and $[0.35, 0.62]$ for academic models. Regression tables can be found in Appendix C.

**Commercial Facial Recognition Models Are Very Accurate**   The commercial models have very high accuracy, particularly AWS and Face++ which each scored above $97.3\%$ accuracy on both verification and identification. As a result, these systems do not have enough incorrect responses to

have any statistically significant conclusions. On the other hand, Azure achieves verification accuracy of $93.3\%$ and identification accuracy of $82.9\%$. In this case, we see a bias towards question gender in favor of males (OR $= 1.76$; $p = 0.041$) which is comparable to the bias observed with humans and academic models.

## 6 DISCUSSION

The study described in this work is the first to compare disparities and bias between humans and machines. We see that the gender and skin type biases of humans are also present in academic models. Interestingly the level of the disparities present in humans are comparable to that of the machines. These human disparities are present even when controlling for the demographics of the participant. We also find that humans perform better when the demographics of the question match their own. It might be easy to look back in hindsight on our results and say they are obvious. It may not be surprising that humans and machines have bias based on gender and skin type. However, we should not forget that this is the first study that directly compares the two with precisely the same questions which are pre-screened for difficulty. Our work is also the first to show a human preferential performance on subjects who look like them.

One key limitation of our human survey is that we analyze a crowdsourced sample. While it is demographically diverse, it does not represent a sample of expert facial recognizers. Our results should not be extrapolated too far outside the sample of non-expert crowd workers located in the US. Additionally, the results we have for the computer models are limited to those which we included and do not represent how all models work or behave.

Our findings contribute meaningfully to the ongoing work of understanding the benefits and harms presented by facial recognition technology. Specifically, we see that automated methods outperform non-expert humans across the board. When bias is detected in a machine, that bias is comparable to those exhibited by non-expert humans. In the future, further work should examine more targeted populations, such as the direct users of facial recognition technology, to understand how their native bias compares to the biases of machines or human-machine teams.

### 6.1 ACTIONABLE INSIGHTS

The field has focused on high-level, aggregate statistics at the dataset level for evaluating model performance. We see that in most areas, including facial recognition, where leader-boards drive innovation and improvements. However, our work puts that into perspective by examining the performance of systems when we look at accurately labeled subgroups of those overall datasets. We also see that commercial models have significantly less bias than academic models or people, and academic models are replicating and have only slightly less bias than people. The first result has never been documented before and leads one to conclude that the commercial companies have expended extra effort and resources to improve the accuracy and decrease the bias in facial identification and verification. This is likely related to incentives to minimize harm and maintain public images; but nevertheless, the different incentive has lead to improvements which academic models don't see because of the improper supremacy of the dataset-level metrics.This paper has actionable contributions to facial recognition for ML Practitioners and Dataset Curators:

**ML Practitioners** (1) Balanced training does not lead to unbiased performance, (2) the CosFace head and ResNet backbones yield lower bias than others, and (3) the ML solutions are always at least as biased as humans. This last point is very important. When we replace a human task with a computer task, obviously speed to do the task is important, but we also want to be better at humans in their biases. Our results, the first direct comparison of humans and machines in this domain, show that this has not been met yet. The models are always as biased, and in many cases, more biased than humans! Thus, we provide a stable benchmark ML practitioners can use to improve upon.

**Dataset Curators** The labels and the means of collecting those labels is very important, and should almost always involve human review. We found that existing datasets' labels (often which used computer-generated labeling methods) were widely insufficient and riddled with errors which have downstream implications on analysis.

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
