# OpenReview forum: "Comparing Human and Machine Bias in Face Recognition"
_ICLR.cc/2023/Conference — Submitted to ICLR 2023_

### Official Review · Reviewer_VUfd · 2022-10-22

**Confidence:** 4
**Correctness:** 2
**Technical Novelty And Significance:** 3
**Empirical Novelty And Significance:** 2
**Recommendation:** 5

**Clarity, Quality, Novelty And Reproducibility:**

The paper is clear and reproducible. There is novelty, but the problem lies in the accuracy of data.

**Details Of Ethics Concerns:**

The paper starts by acknowledging that automatic face recognition has "been criticized in recent years for their intrusive privacy concerns and differential treatment of various demographic groups." It accepts that is a sensitive manner.
The human evaluation has been carried with ethic approval  of the hosting university which is a strong argument for ethic and societal moral fairness.

The paper uses two databases that were criticized for using persons without public consent. Yet in the curation process, the authors " undertook the "task of labeling and categorizing individuals who they do not know and have not received consent from for this task".

Overall, I believe that the paper addresses ethical concern regarding the face recognition as best as possible.

**Strength And Weaknesses:**

Strengths:
 - I totally agree with the paper that there is need in the research to evaluate the bias and compare the human performance with machine performance for the face recognition problem. Contribution here are welcomed
- release of curated database with reference methods evaluated should help the field.
- the paper is well presented and clear

Weaknesses:
1. The main issue of such paper is the following: how it was established the ground truth? By human annotation?! One key aspect of the paper is that it reports a 67%-83% accuracy of human observers accuracy. On other hand the limited number of persons in the authors team provides the ground truth. It is quite hard to take this as an established ground truth especially since there no is strong indication that experts (i.e. authors here) behave much better than normal persons. It is further said several time in the paper, that "one" of the authors revised the data to make sure that no errors are further passed. Yet a person has up to 83% accuracy and quite often 70%. Based on the data provided, curation by machine learning is preferred.
At the end of the paper it is noted: "The labels and the means of collecting those labels is very important, and should almost always involve human review. We found that existing datasets’ labels (often which used computer-generated labeling methods) were widely insufficient and riddled with errors which have downstream implications on analysis." I believe that this is true but is also applicable to the ground truth of this data, since there is no information that consensus of large number of experts was used.

In other words, in my view, the main limitation of the paper is that ground truth is not established accurately. A large number of observers should have been used and average where there is consensus should be placed as labels.

2. The paper does not report the number of incorrect labels in LFW. From the data reported in the paper and in the supplementary material, it seems that the amount of errors was small and most of curation was about removing non-public persons. Unfortunately, in this way the contribution w.r.t the dataset is harder to establish.


**Summary Of The Paper:**

The paper proposes an benchmark (dataset and procedure) to evaluate the bias of human performance in the face recognition problem. To offer more information, the paper reports a set of similar tests on machine performance.

**Summary Of The Review:**

The paper aims to a direction where contributions are welcomed, and there is an auditorium at ICLR. The main  problem is that the accuracy of the data was not well established and in this case is critical.

---

> ### Author Response · Authors · 2022-11-19
> **Response to Reviewer VUfd**
>
> We appreciate your time and careful consideration of our work. We particularly appreciate your concurrence that this work is important and our contribution is welcome. We respond to your most major concerns below.
>
> > Accuracy of the data
>
> We understand that your main concern is in the manner in which we curated the InterRace dataset. We would like to refer you to the Datasheet for Dataset in Appendix E for much more detail on the manner in which our data were collected, due to the space constraints in the main body of the paper.
>
> We understand that your concern is that the “67%-83% accuracy of human observers” calls into question the validity of the human curation of InterRace. We believe that this is a misunderstanding of the result. Table 2 (where we believe you see these numbers) reports the accuracies of humans on face verification and identification tasks. This is not how the dataset was curated. The dataset was not created in a way where a human ever had to perform a verification or identification task. The dataset was created by reviewing and compiling metadata, as well as identifying errors in the CelebA and LFW datasets.
>
> If the concern is about the validity of the survey questionnaire, we suggest that there should be no concerns here as this was a survey that was conducted via computer-generated suggestions of questions and a human review of the question to make sure it wasn’t too obviously easy. The computer-generated questions always ensured that, whether it was a verification or identification question, there was always the proper number of matching identities in the pictures. The human then ensured that the question would not be obvious to either a human or machine survey respondent.
>
> Finally, we note that many gold standard datasets in the machine learning community used human annotators in a process of human annotation just like we did. In fact, when we needed human annotation (like for Fitzpatrick scale annotations), we used 7 annotators, took the majority vote, and broke ties with additional review. This is standard practice in machine learning data curation communities. Additionally, having 7 voters is a large number of human evaluators and we believe provide much stronger assurances of the accuracy of our dataset.
>
> If you have further concerns in light of this clarification, we’d be happy to respond.
>
> > Contribution w.r.t the dataset is harder to establish
>
> Our main contribution is both the InterRace dataset and the survey questions. The dataset provides accurate metadata labels for images in CelebA and LFW on a range of sociodemographic dimensions. Of particular note, skin type metadata are virtually non-existent in face recognition datasets and provide rich and important features to study in such an ethically debatable topic. The questionnaire provides a standard approach to how to measure bias between humans and machines in a fair manner for face identification and verification questions.

---

### Official Review · Reviewer_WUnQ · 2022-10-23

**Confidence:** 5
**Correctness:** 3
**Technical Novelty And Significance:** 2
**Empirical Novelty And Significance:** 2
**Recommendation:** 3

**Clarity, Quality, Novelty And Reproducibility:**

The approach is clearly presented but lacks some details that would make the dataset scrubbing and annotation tasks fully reproducible.  This is a model vs human analysis paper so no training to replicate.  There have been other papers that have compared human to model performance so not completely novel but does present a larger study with more specific tasks than presented prior.

In general, the paper is lacking in quality that meets the standard of the conference as key metrics are missing and results should be more clearly presented and with more detail.

**Strength And Weaknesses:**


The paper is clear and the work is well motivated.  It is certainly important to gauge model performance compared to humans.

It is great that the authors identify the weakness of these face datasets and present cleaned annotations and scrubbing of images that may not be desired in the set.

For some use cases, low quality images may be desired.  It may be preferable to include them but add a quality label to the annotations.  What are the specific thresholds used to decide to omit these images?

"old-timey" should be better defined so as not to be subjectively determined.

If black and white and old-timey images were removed due to the being too easy to recognize as paired then this should be validated but no quantitative check is presented.

"We found that many questions could be answered without considering face features at all, and these
were removed. For example if the subject is wearing identical attire and/or standing in front of an
identical background in two images."  Again, this needs to be shown to be true quantitatively if the authors are referring to the models ability to match faces.  Many models will not use background features making the issue not as important as presented.

"While the Fitzpatrick scale is not perfect, it is the best systematic option currently for ensuring a
broad representation."  Not sure this is still true with the existence of the Monk scale which was specifically created for this task while Fitzpatrick was not, although Fitzpatrick is far more commonly used.

What is the usefulness of the birth date if the date of image capture is not provided?

It would be helpful for future projects if more details on the process and workflow to manually check/verify the annotations in the data were provided.  Lessons learned and the final selected approach would be helpful for other researchers that are interested in a similar process.

It would be much better for the reader if results comparing the models and humans were presented in graphs, figures and tables to provide more detail than what is in Table 2.  This would also allow for high-level take-aways on the details to be better digestible.  A lot of specific statistical results are presented in the text but this should be reflected in figures as well.  Some details can be moved to an appendix if needed.

It would also be helpful to understand the per-human performance.  Some participants may be better at the tasks than others.  The human results would likely form a distribution of performance and this should be captured and compared to the per-model, and academic group as well as commercial group, model distribution.  The uncertainty from these distributions needs to be considered as we are comparing over multiple variables.

Odds ratio is not commonly used by the face rec community so a better description may be helpful

additional metrics such as FMR and FNMR, or FAR and FRR, should be presented.  It would help to understand where the errors are occurring and if the disparities are a result of one type of error or the other.  This is also useful for comparing human to model performance.

Typo at end of section 5: " In this case, we see a bias towards question gender in favor of males"

**Summary Of The Paper:**

The authors aim to answer two questions in this paper, "How and to what extent do humans exhibit
bias in their accuracy in facial recognition tasks?" and "How does this compare to machine learningbased models?"  To address these questions the authors present improvements to the LFW and CelebA dataets as well as present an analysis showing that both humans and models result in lower accuracy on dark skinned and female subjects and humans obtain higher accuracy on the tasks when their demographics match the subjects in the data.  They also find that while academic models are comparable in gender bias compared to humans, they are significantly more biased against darker skin tones than humans.

**Summary Of The Review:**

The paper is clear and the problem is a good one to address but it lacks some key details and presentation limitations that reduce its score.

---

> ### Author Response · Authors · 2022-11-19
> **Response to Reviewer WUnQ**
>
> We appreciate your time and careful consideration of our work. We respond to your most major concerns below.
>
> > There have been other papers that have compared human to model performance so not completely novel
>
> While, yes, there have been other human to machine comparisons before – which we cite in our work, this is the first work that looks directly at biases. It is natural ethical question to ask, whether the types of errors a human would make are similar to the types of errors a machine would make. However, this question has not been asked or answered yet. We provide an answer to that question and find some compelling and concerning results!
>
> > "We found that many questions could be answered without considering face features at all, and these were removed. For example if the subject is wearing identical attire and/or standing in front of an identical background in two images." Again, this needs to be shown to be true quantitatively if the authors are referring to the models ability to match faces. Many models will not use background features making the issue not as important as presented.
>
> We clarify that this quotation was in reference to concerns about human ability, rather than machine ability. We found in our focus groups that if a person was photographed in the same setting, i.e., holding a similar trophy in a jersey, then the human test takers could more easily identify that the individuals were the same people and score higher. We believe that these types of questions would be a potential cofounder in our study and so we removed them. Since it is important to give the humans and machines the same questions, since we removed them for the human test takers, we also removed them from the machine version.
>
> > "While the Fitzpatrick scale is not perfect, it is the best systematic option currently for ensuring a broad representation." Not sure this is still true with the existence of the Monk scale which was specifically created for this task while Fitzpatrick was not, although Fitzpatrick is far more commonly used.
>
> Thank you for the important reference to the Monk scale! We conducted our work with IRB approval before the Monk scale was released in May 2022. We will update the manuscript to include a reference to this.
>
> > It would be helpful for future projects if more details on the process and workflow to manually check/verify the annotations in the data were provided. Lessons learned and the final selected approach would be helpful for other researchers that are interested in a similar process.
>
> We included a Datasheet for Dataset in the Appendix E which we believe contains the type of information desired.

---

### Official Review · Reviewer_TNQq · 2022-10-24

**Confidence:** 4
**Correctness:** 4
**Technical Novelty And Significance:** 2
**Empirical Novelty And Significance:** 3
**Recommendation:** 6

**Clarity, Quality, Novelty And Reproducibility:**

Detailed and well-written. Adding in missing details will help with reproducibility.

**Strength And Weaknesses:**

Strengths:
1. Well-written and structured
2. Experimentally strong
3. Results

Weaknesses:
1. Missing technical novelty.
2. No mention of how the existing model architectures were revised for the task on hand.
3. It is not clear if the apparent bias in data (eg certain age groups were more populated than others) while training/validating and testing.

**Summary Of The Paper:**

Bias in face recognition is a significant area of research that has/will have a meaningful impact on analyzing and managing disparities in downstream applications. The authors of this paper propose enhancements to the existing face datsets such as LFW and CelebA by improving the available metadata and also devise experiments to measure the bias exhibited by humans and models during identification and verification tasks.


**Summary Of The Review:**

Overall, a well-written paper. The findings are interesting but may not be extrapolated as suggested by the authors.

---

> ### Author Response · Authors · 2022-11-19
> **Response to Reviewer TNQq**
>
> We appreciate your time and careful consideration of our work. We respond to your concerns below.
>
> > Missing technical novelty.
>
> Our work is the first to directly compare human and machine bias on face verification/identification. Our main contribution is that architecture has a big influence on fairness, and this can be exploited to discover more fair architectures. We believe that this is impactful towards enabling and encouraging future research on these technologies to ask and answer new and important ethical questions around the adoption of facial recognition.
>
> We agree with you that our paper does not introduce new algorithms, but we respectfully point out that making the raw results from our extensive studies available and establishing a benchmark against human performance is very useful to the face recognition and ethics community. We would also like to mention that experimental-analysis papers have been accepted to recent ICLR, ICML, and NeurIPS conferences, and some of them have public reviews and meta-reviews available here:
>
> [1] Schmidt et al., ICML 2021. [Descending through a Crowded Valley - Benchmarking Deep Learning Optimizers.](http://proceedings.mlr.press/v139/schmidt21a.html)
>
> [2] Ning et al., NeurIPS 2021. [Evaluating Efficient Performance Estimators of Neural Architectures.](https://openreview.net/forum?id=Esd7tGH3Spl&noteId=dl91y63zePD)
>
> [3] Northcutt et al., NeurIPS 2021. [Pervasive Label Errors in Test Sets Destabilize Machine Learning Benchmarks
> .](https://openreview.net/forum?id=XccDXrDNLek)
>
> [4] Mehta et-al, ICLR 2022. [NAS-Bench-Suite: NAS Evaluation is (Now) Surprisingly Easy](https://openreview.net/forum?id=0DLwqQLmqV)
>
>
> > No mention of how the existing model architectures were revised for the task on hand.
>
> We did not change the existing model architectures for this task as our purpose was an extensive, thoughtful, and impactful benchmark of the existing state of the art methods. We outlined the training and algorithmic aspects of the academic models analyzed in Section 4.2.
>
> > It is not clear if the apparent bias in data (eg certain age groups were more populated than others) while training/validating and testing.
>
> We had an entirely balanced training, validation, and test dataset. This allows us to rule out the concern that the training data bias (in terms of balance between the groups) was a contributor to the bias observed in the academic models we tested.

---

### Official Review · Reviewer_NNHX · 2022-11-01

**Confidence:** 2
**Correctness:** 3
**Technical Novelty And Significance:** 1
**Empirical Novelty And Significance:** 2
**Recommendation:** 3

**Clarity, Quality, Novelty And Reproducibility:**

Other than curating the datasets, I do not see much scientific contribution of this work. All the algorithms used for conducting experiments are existing algorithms.

There are several grammatical errors in the paper. Also, it was hard to follow the writing in many sections of the paper.

**Strength And Weaknesses:**

Strengths:
- To the best of my knowledge, this is the first time such a study is done to compare human and machine bias on face verification/identification.
- It looks like the authors have done a good job of curating the datasets by removing poor quality images and correcting the inconsistencies in its metadata.

Weaknesses:
- Other than curating the datasets, I do not see much scientific contribution of this work. All the algorithms used for conducting experiments are existing algorithms.
- The results observed in the paper are expected results. I am not sure how these findings help in further improving the research in the face identification/verification area.


**Summary Of The Paper:**

This paper presents a study that compares disparities and bias between humans and machines in performing face verification and recognition in challenging conditions. The authors use two datasets named LFW and CelebA for these studies. Human experiment consists of randomly sampled subjects answering verification/identification questions and machine experiments were run using existing academic and commercial models.

**Summary Of The Review:**

Overall I think the study presented in paper lacks novelty and the presented results are expected.

---

> ### Author Response · Authors · 2022-11-19
> **Response to Reviewer NNHX**
>
> We appreciate your time and careful consideration of our work. We respond to your concerns below.
>
> > Other than curating the datasets, I do not see much scientific contribution of this work. All the algorithms used for conducting experiments are existing algorithms.
>
> As you agree, our work is the first to directly compare human and machine bias on face verification/identification. Our main contribution is that architecture has a big influence on fairness, and this can be exploited to discover more fair architectures. We believe that this is impactful towards enabling and encouraging future research on these technologies to ask and answer new and important ethical questions around the adoption of facial recognition.
>
> We agree with you that our paper does not introduce new algorithms, but we respectfully point out that making the raw results from our extensive studies available and establishing a benchmark against human performance is very useful to the face recognition and ethics community. We would also like to mention that experimental-analysis papers have been accepted to recent ICLR, ICML, and NeurIPS conferences, and some of them have public reviews and meta-reviews available here:
>
> [1] Schmidt et al., ICML 2021. [Descending through a Crowded Valley - Benchmarking Deep Learning Optimizers.](http://proceedings.mlr.press/v139/schmidt21a.html)
>
> [2] Ning et al., NeurIPS 2021. [Evaluating Efficient Performance Estimators of Neural Architectures.](https://openreview.net/forum?id=Esd7tGH3Spl&noteId=dl91y63zePD)
>
> [3] Northcutt et al., NeurIPS 2021. [Pervasive Label Errors in Test Sets Destabilize Machine Learning Benchmarks
> .](https://openreview.net/forum?id=XccDXrDNLek)
>
> [4] Mehta et-al, ICLR 2022. [NAS-Bench-Suite: NAS Evaluation is (Now) Surprisingly Easy](https://openreview.net/forum?id=0DLwqQLmqV)
>
> > The results observed in the paper are expected results. I am not sure how these findings help in further improving the research in the face identification/verification area.
>
> We agree that some of these results may not be particularly surprising. However, a good result or paper need not surprise the reader. Many previous works, like [GenderShades](http://proceedings.mlr.press/v81/buolamwini18a/buolamwini18a.pdf) in ICML were unsurprising, yet broadly impactful. The documentation of results, particularly in ethically debatable areas like face recognition, allow for further analysis and debate. In our work, by comparing the performance and bias of computers to humans, we can ground debates about the appropriateness of facial recognition in a manner informed by data. Additionally, we conclude our work with two sets of actionable findings which will directly improve the research in face identification/verification. Specifically, we provide insights for face recognition researchers and dataset curators.

---

### Decision · Program_Chairs · 2023-01-20

**Decision:**

Reject

**Justification For Why Not Higher Score:**

AC agrees with the majority of the reviewers.

**Justification For Why Not Lower Score:**

N/A

**Metareview: Summary, Strengths And Weaknesses:**

This paper presents a study on human bias in face recognition that is contrasted with model bias in machine learning. An elaborate human evaluation of two tasks, face identification and verification, has been conducted. A number of conclusions have been reported contrasting machine and human biases in terms of gender, and skin color. AC agrees with the reviewers that albeit it’s an important study, limited technical contribution (how to resolve existing model failures) and a narrow application domain (the paper studies face recognition and bias in face recognition) are two critical issues that place the contributions below the acceptance bar. We hope the reviews are useful to improve the manuscript.